# Developing a stakeholder-informed social responsibility model for translational science

Elise M. R. Smith[1,2,3]*, Georgia Loutrianakis[1,3], Kimberly Beatty[4], Krista Bohn[2], Kathryn A. Cunningham[5], Sharon Croisant[1,2], Jeffrey S. Farroni[1,2,3], Micheal Gienger[6], Dominique Guinn[7], Jometra Hawkins-Sneed[8], Sondip Mathur[9], Victoria McNamara[2], Marnina Miller[10], Stephen Molldrem[1,2,3], Kimberly Pounds[11], Vishnu Subrahmanyam[1,3], Emma Tumilty[1,2,3], Grace A. Loudd[12]

1 School of Public and Population Health, University of Texas Medical Branch, Galveston, Texas, United States of America, 2 Institute for Translational Sciences, Galveston, Texas, United States of America, 3 Institute for Bioethics & Health Humanities, University of Texas Medical Branch, Galveston, Texas, United States of America, 4 Alcohol and Drug Abuse Women's Recovery Center, Galveston, Texas, United States of America, 5 Departments of Pharmacology and Toxicology, Psychiatry and Behavioral Sciences, and Center for Addiction Sciences and Therapeutics, University of Texas Medical Branch Galveston, Texas, United States of America, 6 Galveston Central Church, Galveston, Texas, United States of America, 7 Department of Health, Kinesiology and Sports Studies, Texas Southern University, Houston, Texas, United States of America, 8 Equity Bridge LLC, Houston, Texas, United States of America, 9 Department of Pharmacy Practice & Administration, Texas Southern University, Houston, Texas, United States of America, 10 Positive Women's Network - United States of America, Oakland, California, United States of America, 11 Department of Health Administration & Public Health, Texas Southern University, Houston, Texas, United States of America, 12 Department of Social Work, Texas Southern University, Houston, Texas, United States of America

* elissmit@utmb.edu

## Abstract

Innovation in biomedical research has increased markedly over the last few decades. However, clinical, therapeutic, and public health advances have often not yielded expected improvements in health outcomes nor reduced disparities. Translational science was developed to improve social benefits related to research and development. We propose a practical model for socially responsible translational science that aims to better align research with its expected social benefits. Scientists and community members from the Houston-Galveston region participated in 12 focus groups and a one-day Deliberative Dialogue Summit to examine the expected social benefits of science, establish the factors and practices of social responsibility, and design an empirical model for socially responsible translational science. Researchers and community members discussed three distinct fields of research – HIV, maternal health, and mental health and substance use disorders. We conducted deductive qualitative data analysis based on theoretical social responsibility criteria of translational science, namely: relevance, usability, and sustainability. We then developed inductive codes to capture the factors and practices identified during discussions as necessary for the translation of research to increase social benefit. First, participants explored ways to broaden the scope of biomedical research beyond a narrow emphasis on scientific

**Data availability statement:** Data cannot be shared publicly because deposition would breach compliance with the protocol approved by our research ethics board (also called Institutional Review Board). Given the highly contextualized nature of the study, sharing data would be ethically problematic as it would breach participant privacy regarding sensitive topics like abortion, substance use, and HIV. Sharing data publicly would also breach compliance with the protocol approved by our research ethics board (also called Institutional Review Board. De-identified data will be shared with researchers upon request. Please contact the Iinstitutional Rreview Bboard (irb@UTMB.EDU) to have access to the data set.

**Funding:** All members of this research team were supported by the Clinical and Translational Science Award (UL1TR001439) from the National Center for Advancing Translational Sciences, National Institutes of Health Translational Sciences. (https://ncats.nih.gov/) The funders did not play a role in the study design, data collection and analysis, decision to publish or preparation of the manuscript.

**Competing interests:** The authors have declared that no competing interests exist.

impact to also consider social impacts and determinants of health; this heightens the relevance of research and underscores its responsibility to address social needs and reduce inequities. Second, to improve usability of translational research, participants suggested increasing access to research products, processes, and participation. They also recommended modifying the research infrastructure to incorporate other systems that can assist with translation including the system of care and the broader community-based systems. Third and finally, for the long-term sustainability of research practices, co-development and co-funding of research was promoted to include local community needs, cultures, knowledges and preferences from project commencement to completion.

## Introduction

Significant innovation in biomedical and health research has increased expectations that clinical, therapeutic or public health applications will ensue in a timely manner. However, policy makers highlight the existence of a translational gap often referred to as a 'translation lag' [1] or a 'valley of death' [2] that hinders the application of knowledge in health sciences. American and European policy analysis has linked the translational gap to four factors, namely: the underdeveloped nature of the science of translation which contributes to a lack of technical and epistemic knowledge; insufficient institutional and organization capacities; regulatory barriers; and, fragmentation of the research and innovation ecosystem [3]. Narratives related to translational science have consistently highlighted socio-technical imaginaries wherein biomedical knowledge reduces such barriers and realizes social benefits [4].

In funding calls from the National Center for Advancing Translational Science (NCATS) and a growing body of scholarship, the framing of translational science explicitly emphasizes its responsibility for achieving social benefits of improved health outcomes and reduced health disparities [2,4,5]. NCATS's vision to develop 'more treatments for all people more quickly' is reiterated in their webpages and key documents [6]. The European Infrastructure for Translational Science also explicitly acknowledges that scientific discoveries will 'improve human health and quality of life' resulting in 'better health outcomes for society' [7]. This responsibility has been primarily expressed in terms of the need to reduce regulatory and procedural barriers to speed up innovation, dissemination, and implementation of interventions and, in turn, achieve improved clinical outcomes and equitable population-level benefits. However, responsibility *to society* specifically has not been central in discussions respecting the development of translational science.

Although social responsibility goals of improved health outcomes and reduced disparities are explicit in translational science policies, the notion of social responsibility has been an area of controversy in the scientific literature in research ethics, research integrity and responsible research innovation [8–11]. The American Association for the Advancement of Science (AAAS) report about social responsibility [12] describes how scientists have created a 'culture of responsibility' by agreeing on

consistent internal norms and standards related to certain topics like reproducibility, authorship, mentorship, data management, conflict of interest and protection of animal welfare and human subjects. However, the nature and source of the obligations researchers have towards the broader society remain unclear [11,12].

Researchers and stakeholders more broadly need to understand the social implications of their work if they are to direct their efforts with due consideration of downstream impacts on patient, population, or environmental health. This gap in the literature is consequential in that it undermines researchers' ability to gain public acceptance and achieve successful implementation of translational interventions which depend upon the prerequisite thoughtful and due consideration of, and alignment with, the values, opportunities, and needs of patients and communities.

Social responsibility is particularly important given the recent shift in the United States' (US) landscape where funding agencies are differentiating translational research and translational science. Historically, translational research has often been disease-specific and aimed to reduce specific barriers that exist in various fields. For example, finding a more reliable mouse model that reproduces human phenotypes for a specific type of chemotherapy for pancreatic cancer research can have significant translational research impact. This potential impact is field-specific and study-specific. On a global scale, research funding often follows a disease-specific model, although there is acknowledgment of cross-disease spillover where knowledge from one disease-specific field is used in another disease-specific field [13].

In contrast to translational research within therapeutic areas, translational science seeks to understand the scientific and operational principles that impact the development of sciences in a generalizable manner [14]. This 'disease-agnostic' approach is aligned to less disease-specific and more generalizable goals than those of translational research [14]. For example, creating a mouse model that would closely mimic human immune responses to many different chemotherapies would be an ideal translational science intervention; outcomes could be incorporated into many translational research projects across therapeutic areas.

Translational science aims to modify scientific infrastructures and processes; traditional research ethics norms focused on protecting human subjects no longer suffice because they do not consider systemic issues such as the distribution of social benefit. In previous work, we have argued that extending research ethics beyond its traditional scope to include social responsibility would be more productive to ensure long term beneficial impacts of science [4]. We developed a Social Responsibility Theoretical Framework which demonstrates how the translational criteria of relevance, usability and sustainability apply to promote socially responsible research; the framework could be eventually adapted across therapeutic fields to increase social benefit (improved health outcomes and reduced disparities) [4].

In this study, we further develop and refine the aforementioned Framework based on stakeholder perceptions of social responsibility. This approach is consistent with the scholarship on responsible research innovation in which societal values are integrated into innovation at the outset of, and throughout, the research process and that such values should be deliberated by those affected by research results and innovation [15,16]. Our refinement process yields a social responsibility model which includes practical recommendations intended to apply to future translational science and translational research. Understanding similarities between fields will help identify what factors might be generalizable, system-based and disease-agnostic, while also understanding differences that might point to disease-specific outcomes. Table 1 defines concepts central in this study.

## Methods

### Design and setting

To achieve our goals, we used a pragmatic approach [19,20] to collect empirical data on researchers' and community members' perceptions regarding social responsibility that span across three fields of translational research. We then brought researchers and community members together in a Deliberative Dialogue Summit event to refine our model and understand the commonalities across different fields. In this research, we formed our stakeholder groups from three

**Table 1. Definitions.**

| Translation | "process by which a biomedical observation is turned into an intervention that improves health". [14] |
|---|---|
| Translational research | "the endeavour to traverse a particular step of the translation process for a particular target or disease"[17] |
| Translational science | "investigation which seeks to understand the scientific and operational principles underlying each step of the translational process." [17] |
| Social responsibility in translational science | responsibility for achieving social benefits of improved health outcomes and reduced health disparities |
| Health disparities | "systematic, plausibly avoidable health differences adversely affecting socially disadvantaged groups"[18] |

fields of translational research highly relevant in the local region of Houston-Galveston: 1) HIV research, 2) maternal health research, and 3) mental health and substance use disorder (SUD) research. Using thematic analysis, we analyzed similarities across these fields to develop recommendations based on both researchers' and community members' views (see Fig 1). We used the Consolidated Criteria for Reporting Qualitative Research (COREQ) checklist to ensure rigor and reflexivity [21] (see supporting information S1 File).

This study was reviewed by the Institutional Review Boards at the University of Texas Medical Branch (IRB#23–0034) and Texas Southern University (IRB#1735). The informed consent process started by sending participants an information document that included the purpose of the study, research procedures, risks, benefits, compensation, methods that ensure confidentiality and contact information. Prior to any data collection we explained the main items in the information document and answered any questions from participants. Participants that agreed to participate filled out a short online survey which included their name, demographic information (gender, race and ethnicity) as well as contact information. Verbal informed consent was obtained from all participants in this study; in other words, participants did not sign a document (documentation of written consent was waived). The IRBs approved this consent process.

The three domains of research selected have been areas with important research innovations but with limited translation to equitable societal benefits. For example, in the field of HIV research, daily oral pre-exposure prophylaxis (PrEP) is considered effective; however there is lack of uptake of intervention among populations most affected in Texas [22]. In the case of maternal health, where reproductive technologies have revolutionized the field of research, rates of maternal mortality have paradoxically increased particularly in minoritized communities in Texas [23]. In the field of SUD, Texas is experiencing a major overdose crisis especially in our region of interest; despite the availability of FDA-approved medications, their uptake to improve health in medical practice is very limited [24]. Given the relationship between mental health and SUDs, we decided to include both fields in this study [25].

This study involved the ongoing engagement and collaboration of four community ambassadors who have worked in communities impacted by HIV, maternal health, as well as mental health and SUD. Their involvement in the implementation of this research incorporated invaluable community-based perceptions into its framework, aiming to enhance cultural competency with members of communities that have often been stigmatized by researchers.

The research process consisted of four consecutive stages during 2023–2024:

1. We conducted 12 focus groups to explore the ideals, beliefs and practices related to social responsibility in the three fields of HIV, maternal health, and mental health and SUD research. Half of the focus groups were comprised of researchers only, while the other half consisted solely of community members. Four focus groups with community members took place

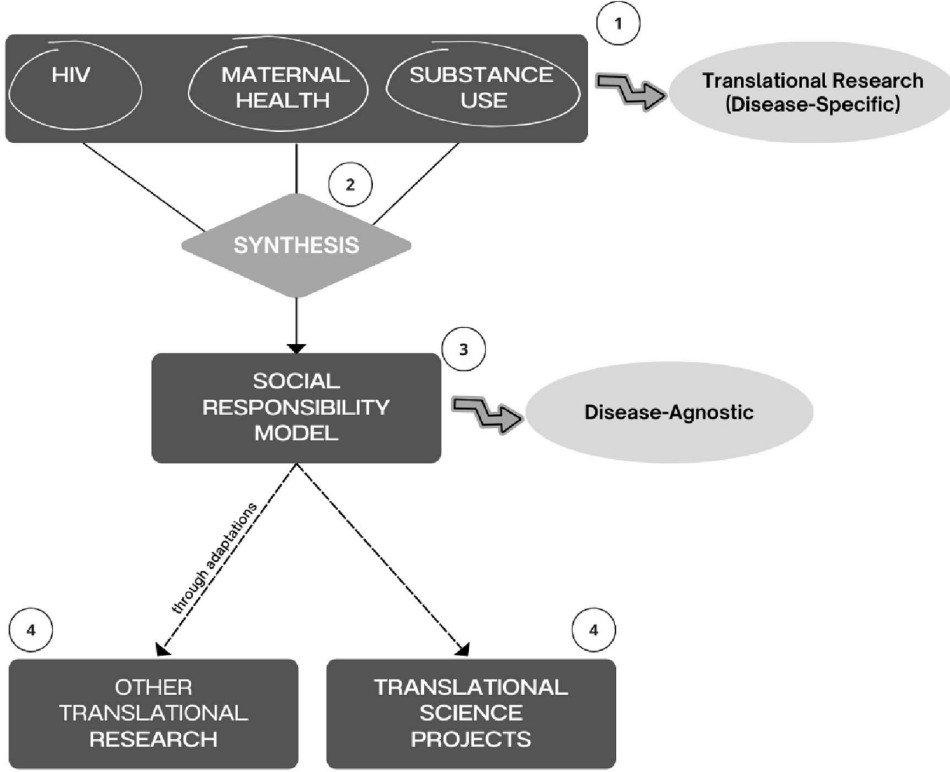

**Fig 1. Overview of the translational science project process.** Process summarizing how knowledge found in many different fields of research (stage 1) can be summarized (stage 2) into a social responsibility model (stage 3) that can be applied in other translational fields or in translational science (stage 4).

in person at community centers with the remaining focus groups with researchers or community members taking place online on Zoom. Focus group guides are available in supporting information documents (S2 and S3).

2. We used translational science social responsibility criteria of relevance, usability, and sustainability to begin deductive coding of transcripts that were further analyzed inductively.

3. We held a Deliberative Dialogue Summit on January 27th, 2024, at the South Shore Harbor in League City, Texas, US. The Summit included several smaller group deliberations influenced by the nominal group technique [26] and culminated with a larger group discussion to explore similarities and differences which emerged from the small group deliberations.

4. We used inductive analysis to refine and modify the social responsibility translational model.

The Deliberative Dialogue approach prioritized knowledge-seeking with the aim to refine and validate practices that promote social responsibility. Although we wanted to identify areas of agreement between stakeholders, our intent was not to direct or force consensus. In designing the Summit, we adhered to three best practices related to Deliberative Dialogues: 1) an appropriate meeting environment, 2) an appropriate mix of participants, and 3) an appropriate use of research evidence [27].

## Participant recruitment

Researchers as well as community members were recruited for the focus groups from the Houston-Galveston region using purposive and snowball sampling. Recruitment started on 10/07/2023 and finished on 26/01/2024. Members of the team identified and emailed potential participant scientists working in domain-specific fields based on various sources of information (e.g., public websites, publications). Community members were selected because they had been impacted directly or indirectly by HIV, maternal health, or mental health and SUDs. Community members included patients, family members, caretakers, non-profit workers and advocates. Recruitment of community members was conducted via email and flyers that were distributed in community centers. Community centers were identified by the community member ambassadors that know where such communities often convene. For the Deliberative Dialogue Summit, we recruited participants from the focus groups and added additional participants using the recruitment process outlined above. Many recruited community members were activists with first-hand knowledge of local social movements. Given the tight knit nature of such community and research groups, a few participants had professional or personal relationships with members of the research team. To our knowledge, this did not seem to hinder discussions. Focus group leaders' backgrounds as well as the reasons why this research was developed was briefly presented at the beginning of focus groups to build rapport with participants.

## Data collection

We conducted two focus groups of researchers and two focus groups of community members in each of the three fields for a total of 12 focus groups. The names of interviewers conducting each focus group as well as their credentials, occupation and sex are available in Table 2. Interviewers had methodological knowledge and expertise in interviewing. Only participants and researchers were present during the focus groups to ensure confidential conversations. Focus group questions were based on a review of the literature on social responsibility by the research team [4]. Core questions asked during the focus groups remained the same for all participants although vocabulary was modified to ensure comprehension. The survey questions were not pilot tested with external participants. We tested the questions within the research team that included both community ambassadors and researchers working in HIV, maternal health and mental health and SUD research. Questions developed for the focus groups are available in the supplementary materials. Probing questions were tailored to the disease-domain. Prior to focus group sessions, we also asked participants to complete a survey using a Redcap application to gather demographic information. Focus groups lasted approximately one hour. No repeat focus groups were carried out with the participants. Brief field notes were made based on experience of interviewers and groups were recorded for analysis.

Detailed field notes were taken during the Deliberative Dialogue Summit to keep a record of conversation throughout the day. Participants were also invited to post any ideas they had on Sticky Notes around the room all of which were recorded as data.

## Data analysis

The focus groups data were recorded, transcribed, anonymized and then validated by members of the team. Three members of the team (Georgia Loutrianakis (GL), Vishnu Subrahmanyam (VS), Elise M. R. Smith (ES)) reviewed the transcripts, uploaded them to the software *Atlas.ti,* and started developing a codebook using an integrative approach to include both deductive and inductive coding [28,29]. Deductive codes were first developed based on translational science social responsibility criteria of relevance, usability and sustainability. Additional codes were developed inductively until coders reached saturation. Two coders (GL, VS) then applied the codes independently to test their application and validate the framework. Coders met weekly to discuss differences and further refined the codebook reiteratively. Lack of consensus generally arose when participants' comments were vague or unclear and therefore might be interpreted in

**Table 2. Name of authors conducting focus groups.**

|  | Interviewers and Credentials | Occupation | Sex |
|---|---|---|---|
| HIV focus groups | Stephen Molldrem, PhD<br>Dominique Guinn, PhD | Assistant Professor<br>Assistant Professor | Male<br>Female |
| Maternal Health focus groups | Kim Pounds, MPH, PhD<br>Georgia Loutrianakis, MA<br>Elise M.R. Smith, MA, PhD | Assistant Professor<br>Graduate Assistant<br>Assistant Professor | Female<br>Female<br>Female |
| Mental health and SUD | Jeffrey S. Farroni, PhD, JD<br>Emma Tumilty, PhD<br>Grace A. Loudd, LMSW, MPA, PhD | Professor<br>Assistant Professor<br>Associate Professor | Male<br>Female<br>Female |

many ways; these comments were not included in results. Once two coders arrived at consensus regarding the presence and placement of codes in four different transcripts, one coder completed the coding of the last eight transcripts. Three members of the team (GL, VS, ES) read through the coding results. The coding results were shared with the full research team which included the community ambassadors. These team members reviewed the coding to ensure accurate reflection of what was observed during focus groups.

Coding results were not returned to individual focus group participants; however, all participants were informed results would be presented at the Deliberative Dialogue and invited to attend as a way for participants to provide feedback on the data and analysis. The Deliberative Dialogue Summit data were coded using the same codebook and coding process. This coding process was an important step to identify commonalities in different fields within one practical model. The same coding framework was applied to all transcripts as the means to identify and summarize similarities. Although differences will be highlighted, they are not central to our model. These field differences will be highlighted in a second manuscript.

## Results

### Overview

A total of 35 scientists and 48 community members were recruited for the focus groups, and a total of 11 scientists and 20 community members participated in the Deliberative Dialogue Summit. The number of researchers per group varied from five to 10 and the number of community members participating in each group varied from three to 10. The range in participant numbers resulted from challenges in recruitment and coordinating meeting arrangements and is acceptable given the exploratory nature of this initial step of the research. Since we selected individuals impacted by particular health issues, it is reasonable that group representation reflects the disparities in each field; however, Hispanic community members were underrepresented in the study. This is important because the Houston-Galveston region demography reflects a substantial Hispanic population that is disproportionately impacted within the three research domains. Additionally, the study was conducted in English only, which means some Hispanic or other ethnic community members whose primary language was not English could not fully participate. Further research would need to be conducted to understand the perspectives of members of the Hispanic and non-English speaking communities. See Table 3 for an overview of demographic participant information. Given the purposive recruitment process it is impossible to know exactly how many participants refused to participate since many simply did not answer the email and we are not sure how many received flyers left at community centers.

Coding of translational science characteristics, relevance, usability, and sustainability were deductively developed based on the social responsibility translational science framework [4]. From focus group discussions of these criteria, we further identified key factors, practices and barriers suggested by different stakeholders. Key factors were often more likely to support certain criteria which is demonstrated in Table 4 and further explained within each result section.

**Table 3. Demographics.**

| Demographics Variable | Focus Groups (N=83) | Deliberative Dialogue (N=31) |
|---|---|---|
| Gender | | |
| Female | 54 | 24 |
| Male | 25 | 7 |
| N/A | 2 | 0 |
| Transgender man | 0 | 0 |
| Transgender woman | 2 | 0 |
| Genderqueer | 0 | 0 |
| I prefer not to answer | 0 | 0 |
| Race | | |
| American Indian/Alaskan Native | 1 | 2 |
| Asian | 6 | 3 |
| Black/African American | 42 | 16 |
| Native Hawaiian or other pacific islander | 0 | 0 |
| More than one racial identity | 4 | 1 |
| N/A | 1 | 0 |
| White | 29 | 9 |
| I do not wish to disclose | 0 | 0 |
| Ethnicity | | |
| Hispanic | 8 | 6 |
| Non-Hispanic | 72 | 25 |
| N/A | 3 | 0 |
| I am Unsure | 0 | 0 |
| I do not wish to disclose | 0 | 0 |

### First criterion of translational science: relevance

**Perceptions from researchers and community members regarding relevance.** We first considered the extent and manner to which research must be relevant in meeting the needs of at least one person, group, or community to effectively impact environmental health, population health or individual health. Scientists highlighted broader notions of relevance than those envisioned by community members. Often scientists looked at population level outcomes in the literature as a measure of the social benefit of their work. Scalability issues were given to explain why certain groups of individuals do not benefit from science.

> ... we are doing a great job of reducing HIV among white, gay men and then all of these other groups, even though the interventions are the exact same, are not benefiting to that degree. So, that is a scalability problem. (…) So, that is a problem that we created by not necessarily thinking about how this is going to impact all populations, instead of just the elite few, per se. (ID: HIV scientists, focus group 1)

According to researchers, studies that lack relevance often lack holistic approaches to health care and do not consider persons throughout their lifespan (e.g., preconception, childhood, post-natal care, menopause). Limitations to the relevance of research were also explained by the exclusion or lack of environmental, familial, intergenerational, and community-based considerations and approaches.

Scientists also recognized that complex disorders or comorbidities are often purposely excluded from research projects to keep samples more heterogeneous. It was acknowledged that homogeneity may oversimplify and diminish the

**Table 4. Overview of participant perceptions, factors and suggested practices.**

| Translational Science Characteristics | Participant | Perceptions | Key factors | Suggested Practices to Enhance Translation |
|---|---|---|---|---|
| **Relevance:** How research is relevant to the needs of at least one person, group or community. | For researchers: | - Being more inclusive<br>- Including more people, broader constructs, and more holistic approaches | - Education<br>- Intersectionality<br>- Inclusion | - Conduct research with a broader scope to include to all people<br>- More data, more populations, better recruitment ideas |
| | For community members: | - Doing research on communities historically excluded from research | - Intersectionality<br>- Fairness and diversity<br>- Education | - Develop research with and for historically excluded populations<br>- Include community partners in projects to educate researchers about needs |
| **Usability:** The usefulness of research, including access, costs of application, and complexity of use. | For researchers: | - Balancing scientific values with social value | - Communication<br>- Transparency of data and results | - Shift the focus of work to include explicit scientific and social values<br>- Incentivize social outcomes |
| | For community members: | - Increased access to knowledge, to participation, to results, and interventions | - Access<br>- Transparency of histories and future collaboration<br>- Bi-directional communication | - Increase community's social responsibility<br>- Increase bi-directional communication to communities so that they can play a bigger role in research |
| **Sustainability:** The long-term use of science, including impact on society and relationships between scientists and community partners. | For researchers: | - How to keep interventions funded after research is done<br>- Politics as a major barrier | - Advocacy<br>- Consistency<br>- Harm reduction | - Develop increased skills to think long term (e.g., dissemination, implementation)<br>- Mentorship – training next generation of researchers with translational skills |
| | For community members: | - Long-term relationships and commitments<br>- Increased and long-term access | - Access<br>- Relational commitments | - Increased collaboration between existing care programs and research<br>- Leverage reiterative infrastructure models to find and maintain sources of funding |

relevance of research. However, scientists also pointed out that large databases may capture relevant data related to complex disorders.

> We need to share that information from thousands of people because of these complex disorders, like substance use disorders and mental health disorders. Why don't we know anything? Because they are so subtle. So, we need thousands and thousands of people to be able to identify changes. (ID: mental health and SUD scientists, focus group 9)

Scientists also highlighted specific barriers to relevance in research. They mentioned that smaller populations that may be exposed to greater risks in research are more reticent, harder to access, and therefore, more challenging to recruit and retain. Lack of funding throughout the pathways from preclinical research to clinical trials was often considered an important barrier in that it can shift the focus to topics less relevant to populations.

Community members perceive the lack of consideration of certain populations at the very outset of research as central in undermining relevance. They often perceive this as a fairness issue where certain groups are excluded, while others are privileged. Community members at the Deliberative Dialogue Summit did recognize an increase in research focused on certain stigmatized populations (e.g., unhoused populations), but they also noted a disappointing lack of results or adequate solutions. Community members in maternal health groups also highlighted that most issues related to maternal

health aim to benefit the fetus, not the mother. This becomes apparent after the child is born and issues linked to postpartum issues may arise. Even if research is conducted, the lack of access to maternal health care makes the translation of research challenging.

> I want to say that a better job of [research] can be done by first acknowledging that women are humans and you can't really make a rule, or a suggestion, or a drug … without understanding that they are real people. They're not less than three-fifths of a person. We're a whole, like, group of people that actually exist and don't really want anyone else in control of our bodies, or how we think about them, or how they're accepted. (ID: maternal health community members, focus group 8)

This previous citation refers to the 1787 US Constitutional Convention in which slaves were counted as three-fifths of a person when determining representation in Congress and taxation. This highlights an ongoing form of racialized violence that has a chilling effect on certain populations and ultimately dampens their interest in participating in future research initiatives.

**Key factors related to relevance.** Key factors related to relevance were inductively identified as often being related to *intersectionality, inclusion, fair representation,* and *education*. Intersectionality can be described as the position of the researcher or community member with regards to race gender, sexuality, demographics and how this position may increase or decrease the impact of research. We refer to intersectionality in its broadest forms as explained by Jennifer Nash [30]. Although scientists did mention broad concepts of inclusion, intersectionality was rarely brought up in a meaningful way. Some did, however, acknowledge how a researcher's positionality differed from that of the communities they study.

> (…) you know, academia is very white. It's very male, and so depending on what groups we're dealing with, and I think especially we're talking about maternal health, we're talking about groups that have historically been vulnerable, continue to be vulnerable in various ways, and that can be hard to build that trust then. (ID: maternal health scientists, focus group 5)

Intersectionality for community members included the understanding of lived experiences which includes the social determinants that impact their life and health. This notion of intersectionality is directly linked to the idea that fair representation will lead to more equitable outcomes. Community members highlighted the need to distinguish similarities as well as differences between groups and to understand the importance of treating all people with respect.

Both researchers and community members were open to further education about social differences and lived experiences. One community member noted how scientists could benefit from a better understanding of the relevance of culture and race in their work. Researchers did often mention that they were not trained to think about the translational impact of their work and how it is relevant to different groups based on sex, gender, culture, and ethnicity. One researcher acknowledged the need for 'retraining and resocialization' (ID: maternal health scientists, focus group 6). Community members also wished for a better understanding of the subject-matter and wished to be informed of upcoming studies in which they could readily participate.

**Recommendations regarding relevance.** The following practical recommendations would improve relevance and could be applied in all fields of health research. First, scientists should learn the detailed demographics of communities relevant to their work to better account for culture, lived experiences and histories, and the social determinants of health and disparities in their work. Scientists can then position their work to have a greater impact leading to generalizability while also addressing the specific needs of different groups. This requires a significant amount of training for some scientists.

Providing educational activities to inform communities about research initiatives underway and under consideration will build relationships and foster the dialogue to engage communities and researchers. To further address lack of knowledge as well as build trust between the community and researchers, community members historically excluded from participation should participate in defining study protocols and ideally play an active role throughout the research process. While cultural competency was seen as a prerequisite in these collaborations, more importantly, community members wanted researchers to clearly demonstrate compassion, empathy and genuine care for communities.

### Second criterion of social responsibility: usability

**Perceptions from researchers and community members regarding usability.**  The second theme, usability, was discussed as related to access, costs of application and complexity of use of the research and downstream research products. This also touched upon the extent to which values, priorities, and preferences of end-users align with scientific development. Scientists admitted that certain topics valued because of statistical scientific significance may not have as much value in terms of their downstream social impact and usability for the community. Scientists pointed out that health issues have biomedical and social components but at times, funders will focus on only one of the two to the detriment of effective implementation of a health intervention.

> So, I think one of the things that has troubled me about HIV for nearly the past decade in terms of funding, has been the swinging of the pendulum, from my perspective, too far in the direction of biomedical research in terms of HIV, right? We all know the history of HIV funding right? Historically, it was very behavioral science focused. (...) Then around 2012 when we had these great studies that came out around treatment, prevention, and PrEP, you know, in around 2015, the pendulum swung really hard the other way towards clinical and uptake of medication, which makes sense, but they did it at the expense of the community. I think that's been the mistake since 2015, right. (ID: HIV scientists, focus group 2)

Scientist did suggest that the most usable interventions would likely be the result of the integration of both biomedical and social components. Broader inclusion of social science fields in health science research was suggested to increase the usefulness of an intervention.

Bench researchers dedicated to working in cellular and/or animal models were much less likely to relate to the long-term useful impacts of their research. This challenge was highlighted when researchers discussed the advancement in mice model research that has only decades later changed therapeutic intervention:

> I think that's why translational research is so challenging, because it's like you're… in the trenches, and you don't feel like you get any wins, and a win will take 10 or 15 years, and you may not even realize you were a part of it because it's like someone else kind of gets the check behind their name, but it took (…) 100 people to get it there. So, anyway, I just wanted to give kudos to them for working on these things that, you know, they never even get credit for. (ID: maternal health scientists, focus group 6)

To make research usable, researchers and community members underscored the importance of co-designing studies. Researchers highlighted how this co-construction of research is a significant shift from more traditional research models that labelled stigmatized groups narrowly as 'vulnerable'. One researcher mentioned that "by labeling maternal and child health or mothers as a vulnerable group, we (…) limit the potentially useful things for this population, and perhaps exacerbate their vulnerability" (ID: maternal health scientists, focus group 6).

The political climate in which research is developed was identified as posing significant challenges to the creation and implementation of usable research. Researchers mentioned that the various scientific stakeholders – including funders

– possess and exercise different types of power and influence that impact the development and potential usefulness of knowledge. And the power dynamics and stakeholder interests at play in the decision-making process are not expected to change significantly:

> Power is a big part, you know, capitalist life, you know, or construction of ideas is a big part. We need money, we need resources, and resources and power are very much tight. So, people on the spectrum of the socioeconomic status are not part of the whole I mean, whether it's, you know, biomedical products are not meant for poor people in so many cases. (ID: HIV scientist, focus group 2)

Beyond the importance of funding in their work, researchers often perceived the political process more broadly as a significant hinderance to the work they do. For example, even when they propose projects that offer promising societal benefits and may have positive state or national political ramifications, they may be rejected outright at the local level by elected representatives who fail to recognize or appreciate what is under consideration.

> So, one of the things that makes me incredibly frustrated in Houston (...) is a lack of value for science and research, right? So, major decisions are made in our local area as well as in the state that's not evidence based and data driven (…) They need to be educated in the value of science. (ID: maternal health scientists, focus group 6)

This connection between science and public policy was mentioned in most focus groups and throughout the Deliberative Dialogue Summit. This is further explored in the sustainability section.

**Key factors related to usability.** When considering usability of research, central factors that facilitate and promote translational science discussed by participants included *access, communication,* and *transparency.* Access touched upon the opportunities or pathways provided to understand and acquire knowledge of research. Access also related to the ability or ease of participants or communities to access and benefit from interventions (e.g., services, medications, and medical devices) resulting from studies, especially if the interventions are not covered by their insurance. Developing lower cost, affordable alternatives which effectively align with the needs of communities was noted by many participants.

Participants also explained how access and transparency are related to bi-directional communication in research. Researchers recognized that raising awareness is part of access and that communication needs to reach and speak to the interests and perceptions of communities. Conversely, researchers in bench science mentioned that communication between community and research is non-existent and even more limited in communities that are underrepresented in science.

> So, there's a huge gap between communication and research. Yes, my community knows that babies are being delivered. They know that we work on COVID, but when it comes to research, animal research, it's like it's really nonexistent (…) There's nobody coming to talk to the younger adults letting them know that research is important. And me, as an African American woman, research was kind of something that just wasn't spoken of. You could go ahead and be a nurse, maybe a doctor, but getting on that bench is very important. (…) I think we could go to the next level by having diversity in research. (ID: maternal health scientists, focus group 6)

Discussions with community members about communication and access to information differed as to what they hoped to obtain about research processes and results. Some community members mentioned that they want as much information as possible at every step of the research, while others suggested that it is more important to provide solutions that harbor a hopeful outlook and avoid disappointing or impractical outcomes.

Researchers and community members discussed how certain types of transparency are central to translation efforts. Researchers saw transparency as study-specific and includes sharing of data, research plans, and knowledge using

different types of dissemination products (e.g., reports, social media, educational materials). A few researchers did mention some limitations to transparency, as in the case of synthesized compounds that could be misused, or in instances in which disclosure of information would threaten the privacy of human subjects. Researchers acknowledged that transparency and communication is an important responsibility but that in practice it could be very complex.

> One of the things that matters to me when I give presentations to the public, it's usually to high school students and to their parents, by the way, I feel a tremendous responsibility to tell the truth. And I think that's really uncomfortable for someone like me who's an addiction researcher. I cannot with a straight face tell people that all drugs are bad at all times after I've had a glass of wine at the reception, you know. So, I just think it's absurd to try to tell a 14 year-old that if they ever, you know, use any drugs their head is going to pop off, their brain is going to explode, and they're going to die. (ID: mental health and SUD scientist, focus group 9)

Compared to researchers, community members saw transparency in much broader terms. They underscored the importance of creating a shared understanding of the past to more fully comprehend historical harms. In their view, transparency implies greater relational commitment to understand community needs and build trust.

> …when I think about social responsibility, I think about transparency, and the transparency that I think about is being transparent with what the goals are, and being transparent with what charity looks like, because charity also includes the time that you spend to actually have conversations, discussions, how much are you actually incorporated into certain organizations. So, I feel like if the goal is set to give voices from individuals or to get their buy in, how can we trust what we are actually giving? How is it bidirectional? How are you being an ally, and how you support me, and how am I supporting you is also important. I think that that's sometimes where the gap is. So, the social responsibility is the transparency. How can we get people to have the clarity they need to be able to make the decisions that they need to benefit themselves and also the other individuals that they support. (ID: HIV community members, focus group 4)

The trust-based relationship is central for the researcher to achieve buy-in and deliver practical benefits to the community. Building trust was not seen as easily achieved and uncomplicated; rather ongoing effort is required to build and sustain the necessary long-term relationships.

**Recommendations regarding usability.** Recommendations to improve usability of the research include designing and conducting interdisciplinary research that factor in both the biomedical and social considerations of health, as well as the active engagement of community members from the onset of research studies and early in the translational pathway. Participants acknowledged that useful research also requires reasonable access to research products. It was suggested that transparency and communication could be improved by introducing partnership programs that bring together researchers and communities. Also, hiring coordinators or researchers with lived experience was deemed very valuable.

On a broader scale, the research infrastructure does not systematically or explicitly consider downstream access issues. Usability of research would be improved through greater collaboration and integration between systems of science and systems of care by prioritizing the needs of care of the community being researched. Participant recommendations for improved access include creating an infrastructure of science that will support downstream access and valuing elements of transparency and communication within community research.

### Third criterion of social responsibility: sustainability

**Perceptions from researchers and community members regarding sustainability.** The third contributing factor to socially responsible translational science is sustainability, defined as the long-term use of science. This may include the development of science that can have a lasting impact on society as well as enduring partnerships that support

sustainable research. Although some researchers viewed long-term uses of research as also involving dissemination and implementation studies, they also noted that they do not have the skills, time, resources or contacts to do so. One researcher mentioned that learning implementation science is 'like teaching an old dog new tricks.' (ID: maternal health scientists, focus group 2)

Sustainability would require researchers to broaden their horizons and the scope of their work which represents a different mindset. To make matters more challenging, researchers at the Deliberative Dialogue did note that what is sustainable for them professionally might not align with what is sustainable for the community. Some researchers perceive their role as developing knowledge within a specific scope which in turn, someone else can use to create practical change. Researchers conducting longitudinal work or bench work highlighted how issues related to sustainability seemed less pertinent to their research. Others thought that promoting open-access publication, data sharing, public dissemination of results to the press or to community groups are all important steps related to promote sustainable science.

Many researchers working with communities highlighted problems with the lack of, or barriers to, the long-term sustainability of interventions in science.

There is, you know, no resource to provide and this is very clear when you look at long acting injectables [to treat or prevent HIV] and even access to PrEP and nPEP, [nPEP refers to Non-Occupational Post-Exposure Prophylaxis] these are very well highly efficacious strategies, but they do not get into the hands of the people who need them for a variety of reasons, right, based on trustworthiness and the ability to do that, but also that whole idea around sustainability and you may have a program that is able to provide long acting injectables for the two years that that grant exist, but then what happens when they pull out and there is no plan to continue because of the way that these things are funded. We know on the research side that for the most part research is disease funded. So, it is based on what the disease of the day, so that often does not translate into sustainable solutions. (ID: HIV scientists, focus group 1)

Researchers mentioned that interrupting access to an intervention which has shown promise and benefited participants may result in harm for those participants and dissuade communities from engaging in future research initiatives. Another significant barrier to sustainable intervention is the system of care, including access to that care. Although researchers distinguish research and care as separate, community members often did not clearly differentiate the two. In fact, discussion of health research with community members would often shift to the lack of resources allocated to health care, even when moderators frequently attempted to focus more narrowly on research. Various groups highlighted the importance of multi-stakeholder approaches in building a truly functional infrastructure for social responsibility, one that would include frontline workers but also those in prevention, care, and monitoring. When discussing how to make research sustainable and useful, communities acknowledged having a role and recognized their own social responsibility in developing research.

One researcher did mention that there is 'siloing of our research and of our healthcare system as a whole' resulting in 'barriers that keep progress from happening in a reasonable time frame' (ID: maternal health scientists, focus group 5). Another researcher mentioned that a great research intervention may be limited when hindered by a problematic infrastructure of care:

You know, you have a plan for what you are going to do, but there is not necessarily resources to be able to do that and often times the solution to some of those is you need to figure out how to integrate this into already existing infrastructures and systems of care, but those systems of care and infrastructures aren't working, which is why you are there to intervene in the first place and so, you have not really necessarily changed the level of resources that they have to be able to sustain something past that. So, I think that the way that the funding mechanisms happen, really does make it challenging to figure out next steps after you determine efficacy. (ID: HIV scientists, focus group 1)

Although community participants agreed that the lack of long-term use of research is problematic, they tended to focus on the need for researchers to collaborate and support already existing programs and infrastructures that could help deliver interventions. Community members further elaborated that integrating knowledge in surrounding infrastructures would include health systems, school systems, non-profit organizations as well as for-profit organizations such as restaurants and hotels. A community member mentioned the need to focus on the quality of community involvement at the outset as well as throughout the life of a longer-term research initiative.

Political and social pressures were also said to have impacted the directions and priorities of long-term research funding. For example, one mental health researcher pointed to the major shift of funding to the opioid crisis, to the detriment of research on medication for methamphetamines and cocaine use disorder.

> You know, for many years I was having great success working on cocaine and methamphetamine, and then the opioid crisis occurred, and I of course take it as seriously as anyone else. But for some reason, that became priority number one, and all of the funding, practically all of the funding has gone towards that. Then everything else was forgotten. Of course those addictions didn't go away. In fact they've worsened over time. It's been really frustrating to try to explain to people why we still don't have a medication for methamphetamine use disorder or cocaine use disorder. (ID: mental health and SUD scientists, focus group 9)

The decreasing number of social programs available locally has only exacerbated the barrier to care, further limiting the impact of research. Communities often mentioned that there is a lack of research as well as the political will and courage to frankly address health inequities linked to general social, political, and economic inequalities.

When discussing how to make research sustainable and useful, communities acknowledged having a role and recognized their own social responsibility in developing research. Various groups highlighted the importance of multi-stakeholder approaches in building a truly functional infrastructure for social responsibility, one that would include frontline workers but also those in prevention, care, and monitoring.

**Key factors related to sustainability.** Practices deemed valuable in promoting sustainable translational science included greater *access* (previously discussed)*, harm reduction, relationship building* and r*elational trust* and *political advocacy*. Regarding harm reduction, both researchers and community members acknowledged that problematic language could increase stigma and negatively impact the reputations of communities. Researchers remarked that stopping an effective treatment and care provided during a study could harm the affected population or communities. Although harm reduction was mentioned as socially responsible translational science, community members were less likely to focus on harm mitigation and preferred to focus on increasing benefits.

Community members mentioned that building relationships with the community at the onset and throughout the research may help in giving stakeholders the time and opportunity to think about long-term funding and the dissemination and implementation of research results. Community members promoted a reiterative process that would enable the continuous flow of information between practice and research. This type of process would also help to effectively identify and draw upon a diversity of funding mechanisms available not only in research but also those supported by non-profits or public services.

> Pairing scientists with community organizations and programs to make sure there is a hand-off I think one of the pieces of this is, and it just kind of struck me, and I think I'm going back to do this, is pairing the research study with the organizations that are doing the work. And after you're able to talk about, here's what we found, here's the demographic, here's the program that we think would be best, here's how we kind of see this, you know, flushing on the community, then pairing up with a nonprofit like Saint Vincent's Clinic [Saint Vincent's is a non-profit clinic that provides essential services to uninsured and underinsured community in Galveston] or other organizations that are doing the work and

saying how could this model help you guys move your work forward, taking what you've seen on paper and actually putting it into action and figuring out ways to make that action work. (ID: maternal health community members, focus group 7)

Although community-research collaboration may open the door to different funding opportunities, authentic collaboration requires a much more significant commitment – one in which scientists learn and respect the knowledge of communities. Community members also pointed out that community advisory boards have historically played a token role within this reiterative process. Acknowledging the place and value of diverse types of knowledge in knowledge translation was deemed as central to achieving sustainable outcomes. Community members suggested relationship building using a lens of historical trauma, especially with black and brown communities. Once there is authentic relationship-building, there can be increased relational trust and agreed upon commitments.

Some community members and researcher did highlight the desire to have researchers influence policy making to allow for more science informed policy. Some senior researchers did recognize that they have a responsibility to advocate politically and institutionally in order to conduct research that is socially responsible; however, they also noted that taking on advocacy might not bode well for junior faculty in a much more precarious position. Some community members did mention the need to leverage the political context to advocate for what they want. One community member introduced the idea of developing a health caucus in collaboration with researchers that would serve as the voice of communities in collaboration with political representatives. The key factors in this section –partnership building, relational trust, political advocacy – work together to build stable infrastructures that help create sustainable benefits.

**Recommendations regarding sustainability.** Prior to the start of a research project, collaboration between researchers and community members or community advocates can serve to further the sustainability of research. A research project may be co-designed with community collaboration so that results are disseminated to create tailored supportive interventions. This type of sustainability can be further expanded through continuous follow-up or dissemination of study results after the end of a project, in the form of result write-ups for the public, published articles, or talks at conferences. Moreover, increased collaboration that links research being done with an already existent care program or institution may also further the translation of research knowledge into sustainable outcomes, even after researchers have left. Based on similarities between groups, concrete recommendations to improve sustainability include: (1) increasing researchers' dissemination and implementation skills; and (2) creating collaborative relationships between research projects and care programs and institutions.

## Discussion

The perceptions of researchers and community members were invaluable in allowing us to explore the social responsibility criteria of relevance, usability and sustainability which had been outlined in a previous theoretical framework. The primary role of stakeholder involvement in this study aimed to align values of research with end-users to increase social benefit. We could not have arrived at the conclusion without the involvement and participation of stakeholders and therefore see this involvement as central to the results. Stakeholder involvement has allowed us to identify key factors and practices and refine our work in building an empirically informed normative model.

The deliberative process in this study allowed participants to understand the reasoning, interests, values and realities of many different stakeholders. Deliberation therefore led to a collective process in which stakeholders had different opinions which were all valued. However, the end-users' opinions were often prioritized since it is only through the application and use of science that benefit is created. Contrary to balancing costs and benefits in an individualistic manner which would align with more classic economic rational choice theory, our social responsibility model is more context dependent and collective in nature. The deliberative process used in this research aligns better with Habermas's theory in which social interaction is central rather than rationality of isolated individuals[31].

We do realize that operationalizing these criteria is a reiterative and dynamic process and that substantial barriers do come into play. Our recommendations are not intended as a 'one size fits all' solution. Rather, our model is more akin to a resource allocation scheme which recognizes the ideal attainment of all goals as simply infeasible and therefore, determines that goals, resources, and practices should be prioritized and directed within specific contexts to optimize the likelihood of achieving benefits. Although we developed the notions relevance, usability and sustainability as separate criteria to facilitate presentation in this article, in practice they should be seen in a dependent manner to avoid silo type thinking. For example, a researcher may develop a new technological innovation that is very relevant to communities since it answers an important need of a population that is underserved. However, the technology might not be useful or sustainable if that same population does not have access to the technology in a sustainable manner that would impact health outcomes and disparities. Further work is needed to refine this type of reiterative process, to find ways to educate stakeholders to take part in this process and to test its efficacy in practice.

Improved health outcomes and reduced health disparities are the main goals of social responsibility in translational science [4]. Researchers seem much more comfortable focusing on having a broad impact on health outcomes rather than reducing disparities which they perceive to be systemic barriers outside their reach. Notably, researchers and community members differed as to the perception, importance and meaning of 'barriers'. Systemic barriers were often used by researchers to justify why they cannot do something. Understandably, with limited time and funding available, working to undo or diminish systemic impediments is no easy feat. Community members did not accept the notion of 'barriers' as a reason not to strive for socially responsible translation and seemed to find alternative ways to proceed. For example, researchers might see the lack of access to health care as a barrier to translation, and, for all intents and purposes, it is a barrier. The differentiation or integration between research and practice has been a topic of significant scholarly discussion since it integrates two different regulatory frameworks [32]. However, community members were quick to leverage community health programs to implement research in a more usable and sustainable way. They demonstrated a fair degree of expertise in pushing against political barriers and advocating for incremental evidence-based changes in policies, activities which are not typically within a researcher's purview. This serves to confirm or validate the value of lay expertise in science production and translation, already acknowledged in the literature more broadly [33].

Community members tended to invariably shift the discussion towards health care implications and seemed to naturally integrate research and practice. Although at first, we believed that this 'drift' was simply a problem of communication and that maybe community members did not understand the scope of research activities discussed, we later realized that by not separating the 'system of science' from the 'system of care,' community members would often argue for the integration of resources from both systems to explore cross-boundary opportunities. This integration of different systems is interesting in that it provides a context for sustainable implementation. Developing metrics to evaluate the relevance, usability and sustainability of this type of implementation could well be a useful next step in evaluating cross-boundary socially responsible translation. Responsible research innovation scholars have highlighted the need to consider the broader context that impacts knowledge production [34]. However, to our knowledge, there have not been any practical recommendations as to how researchers should proceed to do so. Translational researchers have an obligation to think about implementation and dissemination that will be related to the broader social, economic and political structures.

To leverage meaningful lay expertise, participants in our study mentioned that community members need to be accepted as equal partners. Communities do not wish to be labeled as 'vulnerable' and want to be part of research development. Indeed, the social construct of the 'vulnerable human subject' in research ethics regulations has already increased stigma, and contributed to the exclusion of, and the lack of research involving, certain populations [35]. Both community members and scientists highlight how the system has not repositioned itself to correct this and thus has perpetuated this situation. The Institutional Review Board (IRB) still perceives certain groups as systematically vulnerable [36], there is a lack of funding for community engagement [37], and the institutional barriers to such research are significant [38]. A meaningful shift requires that researchers alongside other stakeholders would have to become more reflexive

and effectively adjust their own positioning to align with community members. There already exist many constructive suggestions to provide a more important role to community members, such as: appointing community members to administrative leadership positions; ensuring training on community engagement for clinical researchers and making community engagement obligatory in research focused on historically marginalized communities [39].

We have discussed the merits of a multi-stakeholder model in operationalizing socially responsible research and ultimately contributing to the goals of translational science [4]. However, the move from theory or model to practice will require systemic structural change. Specifically, in order to be effective, the criteria, values, and goals of socially responsible research must not only be broadly communicated, they must also permeate institutions and stakeholder networks. Relevant roles and responsibilities would be distributed at various levels of governance to ensure that socially responsible translation is a collective imperative – a dynamic where the various stakeholders have the space to bring their knowledge to the table and collaborate throughout the research process as well as after, in the delivery of practices that benefit society. Although we limited this specific study to views of researchers and community representatives, future research should include a broader range of stakeholders that influence the system. This collective imperative aligns with the need to shift beyond the typical research triad (researchers, the IRB, and participants) to include public policy makers, industry, non-profits, and communities that may benefit from research [40].

## Conclusion

This research developed a social responsibility model informed by researcher and community perspectives. Results suggest that when discussing what is relevant, researchers think more about broad scientific impact and community members think of local community impact focused on reducing disparities. Although researchers believe that they must balance scientific and social benefits to create usable research, community members put emphasis on transparency of the process and access to research products. Both researchers and community members generally concurred that sustainability had been a shortcoming in many past research experiences. Researchers perceive this to be the result of inadequate funding mechanisms and institutional incentives, and lack of political support. Community members focused more on the lack of relationship building to create long-term sustainable alternatives and the need to diversify funding sources to realize community benefits. In many cases discussed in the deliberative dialogue, researchers and community members could balance short-term and long-term sustainability considerations by looking at different case-by-case examples. In other words, although they focused on different types of sustainability, they could find a shared vision through the deliberation process.

If integrated in policy and practice, this social responsibility model can better align values of scientific innovation with the needs of stakeholders to improve the benefits of scientific development including improved health outcomes and reduced health inequities. Since this model implies individual, team and system level changes which includes a range of stakeholders (researchers, scientific teams, regulators, industry, funding organizations) it would require significant discussion and buy in before implementation occurs. This promotion of social responsibility considering both barriers and practices of translation does focus on practical modifications to the system of science as opposed to an idealized view of scientific innovation.

Future research regarding social responsibility model can include further refining the model with system level stakeholders including funders, industry, regulators, professional societies, community groups and non-profit organizations linked to science translation. A diversity of methods including quantitative and mixed methods would also be beneficial to gain more generalizability. Since translational science implementation and translation will be very different based on the type of health care and public health system, implementation of this model in different countries will require modifications. Studying how different research contexts or demographic regions modify and apply such a model will be a worthy next step to the research on social responsibility in translational science.

Although the institutional shift from translational research to translational science is recent, it remains unclear if and how funding agencies will support the needed modifications to the research infrastructure to ensure increased

downstream social benefit including improved health outcomes and reduced health disparities. In other words, while a range of projects supported by translational science institutions make evidence-based recommendations advocating for a shift to socially responsible infrastructures, there is little in the way of guidance as to how stakeholders would fund, develop, test and implement long-term change.

Aligning collective and individual social responsibilities to promote relevance, usability and sustainability in socially responsible research offers a way forward to support translational science work that will achieve expected goals of increased health outcomes and disparities.

## Supporting information

**S1 File. COREQ checklist.**
(PDF)

**S2 File. Focus group questions for scientists.**
(PDF)

**S3 File. Focus group questions for community members.**
(PDF)

## Acknowledgments

The authors are grateful for the participation of Raven Harding, Tierra Ruth, Tiffany Bystra, and Micheal Kurilla who participated in the Deliberative Dialogue Summit. Authors would also like to acknowledge the important contribution of the research participants.

## Author contributions

**Conceptualization:** Elise M R Smith, Krista Bohn, Kathryn A Cunningham, Sharon Croisant, Jeffrey S Farroni, Micheal Gienger, Dominique Guinn, Sondip Mathur, Victoria McNamara, Stephen Molldrem, Kimberly Pounds, Emma Tumilty, Grace A Loudd.

**Data curation:** Elise M R Smith, Georgia Loutrianakis, Kimberly Beatty, Krista Bohn, Kathryn A Cunningham, Sharon Croisant, Jeffrey S Farroni, Micheal Gienger, Dominique Guinn, Jometra Hawkins-Sneed, Sondip Mathur, Victoria McNamara, Marnina Miller, Stephen Molldrem, Kimberly Pounds, Vishnu Subrahmanyam, Grace A Loudd.

**Formal analysis:** Elise M R Smith, Georgia Loutrianakis, Vishnu Subrahmanyam.

**Funding acquisition:** Elise M R Smith, Jeffrey S Farroni, Dominique Guinn, Victoria McNamara, Stephen Molldrem, Kimberly Pounds, Emma Tumilty, Grace A Loudd.

**Investigation:** Elise M R Smith, Grace A Loudd.

**Methodology:** Elise M R Smith, Kimberly Beatty, Sharon Croisant, Micheal Gienger, Dominique Guinn, Jometra Hawkins-Sneed, Sondip Mathur, Victoria McNamara, Marnina Miller, Stephen Molldrem, Kimberly Pounds, Emma Tumilty, Grace A Loudd.

**Project administration:** Elise M R Smith, Victoria McNamara, Grace A Loudd.

**Supervision:** EliseM R Smith, Grace A Loudd.

**Validation:** Georgia Loutrianakis, Kimberly Beatty, Krista Bohn, Kathryn A Cunningham, Sharon Croisant, Jeffrey S Farroni, Dominique Guinn, Jometra Hawkins-Sneed, Victoria McNamara, Marnina Miller, Stephen Molldrem, Kimberly Pounds, Vishnu Subrahmanyam, Emma Tumilty, Grace A Loudd.

**Visualization:** Georgia Loutrianakis, Sondip Mathur.

**Writing – original draft:** Elise M R Smith.

**Writing – review & editing:** Elise M R Smith, Georgia Loutrianakis, Kimberly Beatty, Krista Bohn, Kathryn A Cunningham, Sharon Croisant, Jeffrey S Farroni, Micheal Gienger, Dominique Guinn, Jometra Hawkins-Sneed, Sondip Mathur, Victoria McNamara, Marnina Miller, Stephen Molldrem, Kimberly Pounds, Vishnu Subrahmanyam, Emma Tumilty, Grace A Loudd.

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
