## [Decision Letter · Decision Letter 0]

20 Nov 2024

Dear Dr. Smith,

Thank you for submitting your manuscript to PLOS ONE. After careful consideration, we feel that it has merit but does not fully meet PLOS ONE’s publication criteria as it currently stands. Therefore, we invite you to submit a revised version of the manuscript that addresses the points raised during the review process.

Please submit your revised manuscript by Jan 04 2025 11:59PM. If you will need more time than this to complete your revisions, please reply to this message or contact the journal office at plosone@plos.org . A rebuttal letter that responds to each point raised by the academic editor and reviewer(s). You should upload this letter as a separate file labeled 'Response to Reviewers'.A marked-up copy of your manuscript that highlights changes made to the original version. You should upload this as a separate file labeled 'Revised Manuscript with Track Changes'.An unmarked version of your revised paper without tracked changes. You should upload this as a separate file labeled 'Manuscript'.

We look forward to receiving your revised manuscript.

Kind regards,

Elizabeth Newnham

Academic Editor

PLOS ONE

Journal Requirements:

2. In the ethics statement in the Methods, you have specified that verbal consent was obtained. Please provide additional details regarding how this consent was documented and witnessed, and state whether this was approved by the IRB

Reviewers' comments:

Reviewer's Responses to Questions

**Comments to the Author**

1. Is the manuscript technically sound, and do the data support the conclusions?

Reviewer #1: Yes

Reviewer #2: Yes

Reviewer #3: Yes

Reviewer #4: Yes

2. Has the statistical analysis been performed appropriately and rigorously?

Reviewer #1: No

Reviewer #2: Yes

Reviewer #3: N/A

Reviewer #4: N/A

3. Have the authors made all data underlying the findings in their manuscript fully available?

Reviewer #1: Yes

Reviewer #2: Yes

Reviewer #3: Yes

Reviewer #4: No

4. Is the manuscript presented in an intelligible fashion and written in standard English?

Reviewer #1: Yes

Reviewer #2: Yes

Reviewer #3: Yes

Reviewer #4: Yes

Reviewer #1: The paper is very interesting and well drafted. The authors may include some statistical information on the subject to corroborate the qualitative findings. Policy implications and usefulness of this paper need to be clearly specified.

Reviewer #2: It’s a well written manuscript that delivers on its stated goals. The introduction reads very well and easy to understand. It provides a brief but sound background to the essence of translational research and contrasted it with translational science. The methods are clear and detailed enough for reproducibility. The results are thoroughly engaging and I quite like the organisation of the results section.

I thought the discussion is good but could be expanded to avoid a silo kind of thinking, something which the authors alluded to briefly under the sustainability criterion. The authors could integrate the three criteria in terms of their dependency. For instance, and depending on how sustainability is conceived, questions of access and transparency (under the usability criterion) could impact on the sustainability of any health initiative.

I also wonder to what extent would those “historical harms” [line 533] play on the minds of some communities thereby influencing the extent of their participation in research (the Tuskegee experiment, and the HeLa cell debacle are two examples that spring to mind.)

Line 594: insert space before the square bracket [...

Great job in my view!

Reviewer #3: Dear Authors, I really appreciated your paper. To improve it, please find my comments below:

1. In the introduction it will be useful to have your definition of social responsibility, equity, disparity, marginalized group. I mean to clarify these concepts and to include a chart that explicit the relations among them.

2. It is not clear to me your reference to the Hispanic group. What is their involvement? Is it the setting you developed the research? or other?

3. in the conclusions suggest to explicit the relation between sustainability int he short and in the long term. Is it a tradeoff or do you see a positive relation, I mean that sustainability in the short term may support the sustainability in the long term.

4. you wrote that it is important to involve all stakeholders. I can see two aspects you should analyze and discuss: how to involve them and how to manage the relations among different stakeholders. I mean how to solve the problem of solutions that have benefits for some stakeholders or lower benefits or costs for others.

5. If I well understand, you wrote that stakeholder involvement is as important as results. Is it correct? If so, how your approach relates to the clinical rationality principle that by definition is related to results? Or do you mean that managing the process is positive related to results?

6. It will be useful to clarify the relation between social responsibility and rational choice. Doses social responsibility mean broadening the concept of rationality?

7. I suggest to better clarify the concept of different group/stakeholders representatives. How are they chosen, for example patient, NOGs, NPOs representatives?

8. It will be useful to propose next steps for this kind of research. Do you see repetition off the research in other context, hopefully international, or do you think it will be netter to analyze some other aspects of the process, or others?

Reviewer #4: This study focuses on a topic relevant to many audiences, stakeholders on research policy, and the research community. The study adheres to rigorous research protocols, and the methodological steps and decisions are clearly described in the paper. The conclusions bring new issues that will add to the ongoing debate on the different understandings of the social responsibility of science. I understand the concerns regarding the anonymity of the participants. Nevertheless, consider the possibility of expending some efforts to anonymize the contents produced by the focus groups to make it available to other researchers.

**Do you want your identity to be public for this peer review?** For information about this choice, including consent withdrawal, please see our Privacy Policy

Reviewer #1: **Yes: ** Dr. JAYANTA KUMAR BASU

Reviewer #2: No

Reviewer #3: **Yes: ** Elio Borgonovi

Reviewer #4: No

---

## [Author Response · Author response to Decision Letter 1]

24 Jan 2025

Response to Reviewers

Journal Requirements:

1. Please ensure that your manuscript meets PLOS ONE's style requirements, including those for file naming. The PLOSONE style templates can be found at

Response: We have reviewed and applied PLOS ONE’s style requirements to the manuscript.

2. In the ethics statement in the Methods, you have specified that verbal consent was obtained. Please provide additional details regarding how this consent was documented and witnessed, and state whether this was approved by the IRB.

Response: This information was added to the ethics statement in the methods: “ To initiate the informed consent process, participants were sent an information document outlining the purpose of the study, research procedures, risks, benefits, compensation, methods that ensure confidentiality and contact information. Prior to data collection, we explained the main items in the information document and answered any questions from participants. Participants who agreed to participate filled out a short online survey which included their name, demographic information (gender, race and ethnicity) as well as contact information. Verbal informed consent was obtained from all participants in this study; in other words, participants did not sign a document (documentation of written consent was waived). The IRBs approved this consent process.”

Response: Data cannot be shared publicly because deposition would breach compliance with the protocol approved by our research ethics boards. Given the highly contextualized nature of the study, sharing data would be ethically problematic as it would breach participant privacy regarding sensitive topics like abortion, substance use, and HIV.

Response: Data cannot be shared publicly because deposition would breach compliance with the protocol approved by our research ethics boards. Given the highly contextualized nature of the study, sharing data would be ethically problematic as it would breach participant privacy regarding sensitive topics like abortion, substance use, and HIV.

Response: The data availability statement has been updated.

Response: Captions have been added below the title Supporting Information.

Response: We have reviewed the reference list. Any additional references were added in response to the comments from reviewers.

Reviewers' comments:

Reviewer's Responses to Questions

Comments to the Author

1. Is the manuscript technically sound, and do the data support the conclusions?

Reviewer #1: Yes

Reviewer #2: Yes

Reviewer #3: Yes

Reviewer #4: Yes

2. Has the statistical analysis been performed appropriately and rigorously?

Reviewer #1: No

Reviewer #2: Yes

Reviewer #3: N/A

Reviewer #4: N/A

Response: This is a qualitative study and therefore there are no statistical analyses.

3. Have the authors made all data underlying the findings in their manuscript fully available?

Reviewer #1: Yes

Reviewer #2: Yes

Reviewer #3: Yes

Reviewer #4: No

Response: Data cannot be shared publicly because deposition would breach compliance with the protocol approved by our research ethics boards. Given the highly contextualized nature of the study, sharing data would be ethically problematic as it would breach participant privacy regarding sensitive topics like abortion, substance use, and HIV.

4. Is the manuscript presented in an intelligible fashion and written in standard English?

Reviewer #1: Yes

Reviewer #2: Yes

Reviewer #3: Yes

Reviewer #4: Yes

5. Review Comments to the Author

Reviewer #1:

Comment: The paper is very interesting and well drafted. The authors may include some statistical information on the subject to corroborate the qualitative findings. Policy implications and usefulness of this paper need to be clearly specified.

Response: The goal of this specific paper is to develop a practical model for socially responsible translational science based on rich multidisciplinary discussion with diverse stakeholders. Given the complexity of the qualitative data, we decided to limit this manuscript to a qualitative analysis. Including statistical information in this study does not align with the methodological approach and would detract from the goal of this research. However, we have added in the conclusion (p. 41-42) that future studies using a diversity of methods including quantitate approaches should be developed. Policy implications and usefulness of this paper have been added in the discussion and conclusion sections of the paper (p.41).

Reviewer #2:

Comment: It’s a well written manuscript that delivers on its stated goals. The introduction reads very well and easy to understand. It provides a brief but sound background to the essence of translational research and contrasted it with translational science. The methods are clear and detailed enough for reproducibility. The results are thoroughly engaging and I quite like the organization of the results section.

I thought the discussion is good but could be expanded to avoid a silo kind of thinking, something which the authors alluded to briefly under the sustainability criterion. The authors could integrate the three criteria in terms of their dependency. For instance, and depending on how sustainability is conceived, questions of access and transparency (under the usability criterion) could impact on the sustainability of any health initiative.

Response: We thank the reviewer for this great suggestion. We have added information regarding the dependency of principles in the discussion section on p.37. “Although we developed the notions relevance, usability and sustainability as distinct criteria to facilitate our analysis in this article, in practice they are inter dependent and must be seen as such. For example, a researcher may develop a technological innovation that is very relevant in that it responds to the needs of an underserved population. However, that innovation might not be useful or sustainable if the population does not have access in a sustainable and timely manner to effectively impact health outcomes and disparities.”

Comment: I also wonder to what extent would those “historical harms” [line 533] play on the minds of some communities thereby influencing the extent of their participation in research (the Tuskegee experiment, and the HeLa cell debacle are two examples that spring to mind.)

Response: As mentioned on p.27, when discussing “historical harms”, community members wanted to discuss past harms openly to avoid making such mistakes again. We did not go into how this impacts participation in research in this study and therefore details of this topic are outside of the scope of this research. However, the topic is already very well studied in the scholarly literature including the following articles:

• Scharff, Darcell P., Katherine J. Mathews, Pamela Jackson, Jonathan Hoffsuemmer, Emeobong Martin, and Dorothy Edwards. 2010. “More than Tuskegee: Understanding Mistrust about Research Participation.” Journal of Health Care for the Poor and Underserved 21 (3). Johns Hopkins University Press: 879–897. https://muse.jhu.edu/pub/1/article/389044.

• Dhai, A. 2017. “Exploitation of the Vulnerable in Research: Responses to Lessons Learnt in History.” South African Medical Journal 107 (6): 472–474. doi:10.7196/SAMJ.2017.v107i6.12437.

Comment: Line 594: insert space before the square bracket [...Great job in my view!

Response: space has been added.

Reviewer #3:

Comment: Dear Authors, I really appreciated your paper. To improve it, please find my comments below:

1. In the introduction it will be useful to have your definition of social responsibility, equity, disparity, marginalized group. I mean to clarify these concepts and to include a chart that explicit the relations among them.

Response: Table has been added on p.7 with the concepts that are central in the introduction and throughout the paper.

Comment: 2. It is not clear to me your reference to the Hispanic group. What is their involvement? Is it the setting you developed the research? or other?

Response: Yes, it is the research setting that establishes a link to the Hispanic population. We added information to page 14 to further describe the setting and clarify the reference.

3. in the conclusions suggest to explicit the relation between sustainability in the short and in the long term. Is it a tradeoff or do you see a positive relation, I mean that sustainability in the short term may support the sustainability in the long term.

Response: We have explicitly added an explanation at the end of the first paragraph of the conclusion (p.41) to clarify the dynamic between short term and long term sustainability considerations.

4. you wrote that it is important to involve all stakeholders. I can see two aspects you should analyze and discuss: how to involve them and how to manage the relations among different stakeholders. I mean how to solve the problem of solutions that have benefits for some stakeholders or lower benefits or costs for others.

Response: This has been clarified in the discussion on p. 36: “The perceptions of researchers and community members were invaluable in allowing us to explore the social responsibility criteria of relevance, usability and sustainability which had been outlined in a previous theoretical framework. Stakeholder involvement in this study was crucial for us to better align values of research with end-users to increase social benefit. The active and constructive participation of stakeholders allowed us to identify key factors and practices and refine our work in building an empirically informed normative model.

The deliberative process applied in this study allowed participants to engage and exchange views to gain a better understanding of their diverse perspectives, reasoning, interests, values and realities. This led to a collective and constructive dialogue where different opinions could be valued. Notably, the end-users’ opinions were often prioritized since it is only through the application and use of science that benefit is created.”

5. If I well understand, you wrote that stakeholder involvement is as important as results. Is it correct? If so, how your approach relates to the clinical rationality principle that by definition is related to results? Or do you mean that managing the process is positive related to results?

Response: We clarified in the discussion: “Stakeholder involvement in this study was crucial for us to better align values of research with end-users to increase social benefit. The active and constructive participation of stakeholders allowed us to identify key factors and practices and refine our work in building an empirically informed normative model.” Stakeholder involvement is certainly also valuable to gain stakeholder buy-in, understanding and collaboration. We never state that stakeholder involvement is more or less important than results of our study. However, since we could not have arrived at our results without stakeholder involvement, one may conclude that it is as important.

6. It will be useful to clarify the relation between social responsibility and rational choice. Does social responsibility mean broadening the concept of rationality?

Response: we have compared rational choice with social responsibility on p. 36.

7. I suggest to better clarify the concept of different group/stakeholders representatives. How are they chosen, for example patient, NOGs, NPOs representatives?

Response: we clarified the concept of community representative on p. 11 in the recruitment section. We also clarified the need to include a broader number of stakeholders at the end of the discussion on p.40.

8. It will be useful to propose next steps for this kind of research. Do you see repetition off the research in other context, hopefully international, or do you think it will be netter to analyze some other aspects of the process, or others?

Response: Next steps for this kind of research has been added in the conclusion on p. 41-42.

Reviewer #4: This study focuses on a topic relevant to many audiences, stakeholders on research policy, and the research community. The study adheres to rigorous research protocols, and the methodological steps and decisions are clearly described in the paper. The conclusions bring new issues that will add to the ongoing debate on the different understandings of the social responsibility of science. I under

---

## [Decision Letter · Decision Letter 1]

27 Feb 2025

Developing a Stakeholder-Informed Social Responsibility Model for Translational Science

PONE-D-24-42697R1

Dear Dr. Smith,

We’re pleased to inform you that your manuscript has been judged scientifically suitable for publication and will be formally accepted for publication once it meets all outstanding technical requirements.

Kind regards,

Emily Lund

Academic Editor

PLOS ONE

Additional Editor Comments (optional):

Reviewers' comments:

Reviewer's Responses to Questions

**Comments to the Author**

Reviewer #1: All comments have been addressed

Reviewer #2: All comments have been addressed

Reviewer #3: All comments have been addressed

Reviewer #4: All comments have been addressed

2. Is the manuscript technically sound, and do the data support the conclusions?

Reviewer #1: Yes

Reviewer #2: Yes

Reviewer #3: Yes

Reviewer #4: Yes

3. Has the statistical analysis been performed appropriately and rigorously?

Reviewer #1: N/A

Reviewer #2: Yes

Reviewer #3: Yes

Reviewer #4: N/A

4. Have the authors made all data underlying the findings in their manuscript fully available?

Reviewer #1: Yes

Reviewer #2: Yes

Reviewer #3: Yes

Reviewer #4: Yes

5. Is the manuscript presented in an intelligible fashion and written in standard English?

Reviewer #1: Yes

Reviewer #2: Yes

Reviewer #3: Yes

Reviewer #4: Yes

Reviewer #1: The author has taken care of the comments. Thae author has also tried to defend some of the comments constructively.

Reviewer #2: Line 176: May need to remove the figure 1 caption here as there is no such caption for figure 2 in the main text.

Reviewer #3: Dear authors, thanks for your modifications. I really appreciated your efforts in making all changes required.

Reviewer #4: (No Response)

**Do you want your identity to be public for this peer review?** For information about this choice, including consent withdrawal, please see our Privacy Policy

Reviewer #1: **Yes: ** Dr Jayanta Kumar Basu

Reviewer #2: No

Reviewer #3: No

Reviewer #4: **Yes: ** Elizabeth Balbachevsky

---

## [Editor Report · Acceptance letter]

PONE-D-24-42697R1

PLOS ONE

Dear Dr. Smith,

I'm pleased to inform you that your manuscript has been deemed suitable for publication in PLOS ONE. Congratulations! Your manuscript is now being handed over to our production team.

Kind regards,

on behalf of

Dr. Emily Lund

Academic Editor

PLOS ONE